# Cine-MRI and T1TSE Sequence for Mediastinal Mass

**DOI:** 10.3390/cancers16183162

**Published:** 2024-09-15

**Authors:** Matthias Grott, Nabil Khan, Martin E. Eichhorn, Claus Peter Heussel, Hauke Winter, Monika Eichinger

**Affiliations:** 1Department of Thoracic Surgery, Thoraxklinik, Heidelberg University Hospital, Roentgenstrasse 1, 69126 Heidelberg, Germany; 2Translational Lung Research Center Heidelberg (TLRC-H), German Center for Lung Research (DZL), Im Neuenheimer Feld 130.3, 69126 Heidelberg, Germany; 3Department of Diagnostic and Interventional Radiology with Nuclear Medicine, Thoraxklinik, Heidelberg University Hospital, Roentgenstrasse 1, 69126 Heidelberg, Germany; 4Department of Diagnostic and Interventional Radiology, University Hospital Heidelberg, Im Neuenheimer Feld 420, 69120 Heidelberg, Germany

**Keywords:** surgical resection, mediastinal mass, cine-MRI, computed tomography, T1TSE sequence

## Abstract

**Simple Summary:**

Contrast-enhanced computed tomography (CT) is the standard radiologic examination for evaluating mediastinal tumors. In equivocal cases, cine magnetic resonance imaging (cine-MRI) and magnetic resonance imaging (MRI)/T1-weighted spin echo sequences (T1TSE) may be additionally performed. We retrospectively analyzed patients undergoing surgical resection of mediastinal tumors (benign and malignant) with prior CT, cine-MRI, and T1TSE for (re-)evaluation and comparison to the intraoperative findings and postoperative histology reports, which were defined as a gold standard. Unclear CT cases were further investigated. A total of 47 patients were included in the study. Cine-MRI is of crucial benefit in unclear CT findings compared with T1TSE, especially when infiltration into the large central vessels and atria is suspected.

**Abstract:**

**Background/Objectives:** Contrast-enhanced computed tomography (CT) is the standard radiologic examination for evaluating the extent of mediastinal tumors. If tumor infiltration into the large central thoracic vessels, the pericardium, or the myocardium is suspected, cine magnetic resonance imaging (cine-MRI) can provide additional valuable information. **Methods:** We conducted a retrospective study of patients with mediastinal tumors who were staged with CT, cine-MRI, and a T1-weighted turbo spin echo (T1TSE) prior to surgical resection. Imaging was re-evaluated regarding tumor infiltration into the pericardium, myocardium, superior vena cava, aorta, pulmonary arteries, and atria and compared with intraoperative findings and postoperative histopathological reports (gold standard). Unclear CT findings were further investigated. **Results:** Forty-seven patients (29 female and 18 male patients; median age: 58 years) met the inclusion criteria. Cine-MRI was able to predict infiltration of the aorta in 86%, pulmonary arteries in 85%, and atria in 80% of unclear CT cases. Aortic tumor infiltration in unclear CT cases was significantly more often correctly diagnosed with cine-MRI than with T1TSE sequence. **Conclusions:** Additional cine-MRI is of crucial benefit in unclear CT cases. We recommend performing cine-MRI if infiltration into the large central vessels and atria is suspected. T1TSE sequence is of very limited additional value.

## 1. Introduction

Contrast-enhanced non-cardiac gated computed tomography (CT) of the chest is the imaging modality of choice for assessing anterior mediastinal masses and their relation to the large intrathoracic vessels, the pericardium, and myocardium and is sufficient for the staging of thymic epithelial tumors [1]. Standard report terms for CT (reports) exist and were approved and adopted by the International Thymic Malignancy Interest Group (ITMIG) members in February 2011 [2]. Contour and adjacent lung abnormalities in CTs correlate with the Masaoka–Koga and World Health Organization (WHO) histological classifications in thymoma patients [3,4]. Some CT imaging criteria such as loss of intervening fat planes, a wide contact area and/or large contact angle (90° or 180°), indentation or distortion of the vessel contour, and intraluminal extension are suggestive of tumor infiltration of adjacent (vascular) structures [5,6].

In these doubtful cases, (static) magnetic resonance imaging (MRI) can help [7]. MRI offers higher soft-tissue contrast with better tissue characterization than CT [8]; there is no radiation exposure with MRI [9]; MRI may be indicated instead of CT in patients with iodine contrast allergy or renal insufficiency instead of CT [10,11,12] or in pediatric patients [13]; MRI is better suited to differentiating thymic hyperplasia from thymoma or when a cystic (thymic/pericardial) lesion requires further investigation [14,15]; and tissue characterization by MRI is superior to CT [16].

One possible MRI technique for assessing tumor infiltration is chemical shift imaging (CSI). CSI is sensitive for detecting microscopic fat [17]; CSI is useful for differentiating thymic hyperplasia from thymic neoplasm and thymic tumors [18], as thymic hyperplasia shows a homogenous decrease in signal intensity of opposed-phase sequences compared to in-phase sequences [19]. Sensitivity, specificity, and accuracy of CSI for confirming or excluding intrathoracic vascular tumor infiltration are 75.7%, 97%, and 91.2%, respectively [5]; and CSI is superior to CT [20,21].

Another MRI technique is subtracting MRI (post–pre-contrast), which can help to detect solid enhancement in the evaluation of cystic mediastinal lesions [8]. Diffusion-weighted imaging is another technique for further clarification of mediastinal masses. Water diffusion is restricted in high-cellularity tissues like tumors. This leads to a higher signal intensity in diffusion-weighted MRI [22].

Static MRI is not the tool of choice for assessing lung infiltration by mediastinal tumors. The lung tissue lacks protons, resulting in poor MRI signal return, thus reducing the evaluation of lung involvement [6,23]. For the assessment of lung infiltration, CT is more suitable due to its higher spatial resolution. 18F-FDG PET/CT is not routinely performed to further evaluate or characterize a mediastinal mass [14].

Cinematic steady-state free-precession (SSFP) MRI is available for assessing the sliding motion between the tumor and the adjacent structures (henceforth referred to as “cine-MRI”) [6,9,17,24,25].

Heart pulsation and respiration limit the demarcation of those structures; thus electrocardiogram (ECG) gating/triggering is necessary for high spatial imaging without pulsation artifacts, together with breath-hold imaging to achieve a good spatial resolution. Eliminating pulsation artifacts by ECG gating would be the main advantage of cine-MRI over static MRI and CT. In addition to excellent soft tissue contrast, real-time imaging delivers visual mechanical information about the adherence of the structures during the cardiac cycle. Additional MRI T1-weighted spin echo sequences (T1TSE) may be valuable for evaluating intervening fat planes [17,26]. A slice thickness of ≤5 mm for cine-MRI is recommended by the Korean Society of Thoracic Radiology [22].

Cine-MRI might help to evaluate the extent of tumor infiltration prior to surgical resection in advanced thymoma [27]. Previous studies demonstrated—in unequivocal CT cases—a sensitivity of cine-MRI for tumor infiltration in adjacent structures (such as pericardium, superior vena cava (SVC), or aorta) ranging from 50% to 88.5% [5,6,28]. As complete surgical resection for thymic tumors is the aim of any surgical treatment, due to its impact on the patient’s survival [29], cine-MRI may be of vital importance in the treatment strategy.

It is generally accepted and has been demonstrated in some studies that cine-MRI provides an additional benefit in the assessment of mediastinal masses when CT findings are unequivocal. The literature on the use of cine-MRI in the assessment of mediastinal masses is still limited [6]. The additional benefit of the T1TSE sequence on cine-MRI is unknown. In daily clinical routine, we have the impression that cine-MRI is of additional benefit compared to the T1TSE sequence in these cases.

We therefore conducted a retrospective study in our clinic to evaluate CT, cine-MRI, and T1TSE sequences of patients prior to surgical resection. The aim was to compare the accuracy of the imaging and to formulate recommendations for clinical routine. Unclear CT cases and the additional benefit of T1TSE sequences were further investigated.

## 2. Materials and Methods

### 2.1. Patient Selection

Following the approval of the Ethical Board Committee of the Medical Faculty of Heidelberg, our clinical (EPOS, SAP, Walldorf, Germany) and radiological (Synapse5, FujiFilm, Minato, Tokyo, Japan) documentation systems were searched for patients undergoing surgical resection for a solid mediastinal mass. Preoperative cine-MRI, T1TSE sequence, and CT had to be available for (re-)evaluation. The interval between techniques must not have exceeded three months to avoid confounding results from (possible) tumor progression. We arbitrarily chose three months as a cutoff (independent from mediastinal mass histology) according to the ESMO guideline on imaging follow-up of thymic epithelial tumors, which is performed in three-month intervals [1]. Data on intraoperative findings and postoperative histopathology reports were obtained from patients’ records and defined as the gold standard.

### 2.2. Inclusion Criteria

Inclusion in the study was independent of histological tumor subtype, surgical approach, or neoadjuvant treatment (chemotherapy and/or radiotherapy). Absence of distant/extrathoracic metastases and a patient age above 18 years were mandatory for inclusion. The study enrolled patients who were scheduled for both primary resection and reoperation for disease recurrence. Surgical approaches included anterolateral (aTHCT), posterolateral (pTHCT), double (dTHCT), clamshell (cTHCT), and hemiclamshell (hcTHCT) thoracotomy, as well as median sternotomy (mST), video-assisted thoracic surgery (VATS), and robotic-assisted thoracic surgery (RATS).

### 2.3. Surgical Course

The goal for every patient was to achieve a macroscopic complete resection (R0/R1). Radical ‘en-bloc’ resection of the tumor with the surrounding perithymic fat (in case of thymic epithelial tumors) and/or adjacent structures, including the lung, the pericardium, the SVC, or the nerves (phrenic nerve, recurrent laryngeal nerve), was performed when necessary. In patients with pleural metastases, the surgical procedure was extended to pleurectomy +/− hyperthermic intrathoracic chemotherapy (HITOC) in indicated cases. Replacement of resected SVC was conducted using a (ring-augmented) polytetrafluorethylene (PTFE) prosthesis. The pericardium was reconstructed with bovine pericardial patches.

### 2.4. Cine-MRI and CT Protocols

cine-MRI has been available at our clinic since 07/2012 using automated breath-holding instructions. Examinations were performed on a 1.5 Tesla MRI scanner (Magnetom Aera, Siemens, Erlangen, Germany) using an 18-channel phased array coil. It was performed in inspiratory breath-hold with a steady-state free precession (SSFP) sequence, field of view of 550 × 450 mm, repetition time (TR) of 45 ms, echo time (TE) of 1.24 ms, acquisition time (TA) of 5.62 × 10 ms per image, and individually stacked slices of 5 mm thickness. T1TSE without fat suppression was performed with a T1-weighted turbo spin echo sequence, field of view of 550 × 450 mm, repetition time (TR) of 799 ms, echo time (TE) of 23 ms, acquisition time (TA) of 6.39 × 8 ms per image, and slice thickness of 5 mm, with ECG triggering in inspiratory breath-hold. Both sequences were performed in a transversal plane and a second plane individually angulated perpendicular to the suspected infiltration between the mass and the organ in question. Image interpretation reported the angle of circumferential contact to the organ in question, the angle of lacking fat interposition, and the area of mechanical adherence (modified according to Marom et al. [2]). Diffusion restriction and additional relevant findings were documented, if available.

CT was often performed out-of-house and included (iodine-) contrast enhancement with 1 mm to 5 mm slice thickness. Cine-MRI, T1TSE sequences, and CT images were reassessed by dedicated thoracic radiologists (ME, 20 years of experience in thoracic radiology). To reduce selection bias, the radiologist was not aware of the type of histology or the surgical approach. Tumor extension was assessed for infiltration of the following structures in cine-MRI, T1TSE sequence, and CT: pericardium, myocardium, SVC, aorta, pulmonary arteries, and atria. Imaging evaluation choices for tumor infiltration included “yes”, “no”, and “unclear” and were gathered in a score sheet for each patient. “yes” and “no” were the only options for the gold standard (Appendix A).

### 2.5. Statistical Analysis

Obtained score sheets were compared to the intraoperative findings and postoperative histopathology reports [28]. For each structure, a true positive and true negative event was registered when the radiological assessment and the gold standard matched. In case of a mismatch, a false positive or false negative event was noted. Subsequently, positive predictive value (PPV), negative predictive value (NPV), sensitivity, and specificity as well as accuracy and Dice similarity coefficient (DSC) for each structure and imaging modality (cine-MRI, T1TSE sequence, and CT) were calculated. The calculation was “Zero” if the numerator gave the result “Zero” (e.g., if there were no false positive events after evaluation). If the denominator gave the result “Zero” (e.g., no true positive and false positive events), the calculation was marked “not possible”. We defined a value of >0.95 as “excellent”, >0.9 as “good”, and <0.9 as “poor”. Two-tailed Fisher’s exact test—due to the low sample size and for categorical data—was used to compare cine-MRI with T1TSE sequence results in unclear CT diagnoses. A *p*-value of <0.05 was considered statistically significant.

## 3. Results

### 3.1. Patient Selection and Characteristics

Between 02/2013 and 01/2022, a total of 107 patients underwent CT, cine-MRI, and T1TSE sequences for mediastinal mass evaluation prior to surgery. A total of 53 patients were excluded due to incomplete or missing data, five patients due to the interval between imaging modalities exceeding three months, and two patients due to surgery for biopsy only (no surgical tumor resection), leaving 47 patients eligible for evaluation (Figure 1). Patient characteristics, median tumor diameter (determined in the final histology report), and surgical approaches are shown in Table 1 and the final histology reports are in Table 2.

### 3.2. Primary Endpoint: Calculation of Statistical Tests for Each Imaging Modality

In summary, primary endpoints (sensitivity, specificity, PPV, NPV, accuracy, DSC) depict acceptable radiological assessment to rule out tumor infiltration with all three imaging modalities for all structures except the pericardium. Table 3 gives an overview of the calculated primary endpoints. Appendix A depict true positive and true negative events counted for each imaging modality. Appendix A, show a patient with a false positive CT but a true positive T1TSE sequence confirmed surgically.

### 3.3. Secondary Endpoint: Unclear CT Cases

Unclear CT cases were further checked for the correct diagnosis in cine-MRI and T1TSE sequences. Unclear aortic infiltration in CT is correctly diagnosed by cine-MRI and T1TSE sequence in 86% and 29% of the cases, respectively. Equivocal pulmonary artery infiltration in CT is correctly diagnosed by cine-MRI in 85% of the cases compared to 54% of the cases by T1TSE sequence. In 80% of cases, atrial infiltration is correct in cine-MRI assessment but only 20% in T1TSE sequence. Both imaging modalities (cine-MRI and T1TSE sequence) were not able to make the right diagnosis in unclear CT cases when assessing tumor infiltration of the SVC and were correct in only 50% of patients with myocardial infiltration (annotating the low number of patients with SVC and myocardium).

Fisher’s exact test confirmed a significant difference between cine-MRI and T1TSE sequence for aortic tumor infiltration (*p* = 0.0004), favoring cine-MRI. For all other structures, a statistically significant difference could not be demonstrated (Table 4).

When further looking at the unclear CT cases which were also incorrect (unclear, false positive, false negative) in cine-MRI assessment, the added value of the T1TSE sequence was very limited. In only a maximum of 50% of the patients was T1TSE sequence assessment correct.

Appendix A show a patient with an unclear CT diagnosis for aortic infiltration, but a correct cine-MRI diagnosis (no infiltration apparent, true negative). Unclear T1TSE sequence was confirmed intraoperatively and in the final histology report.

## 4. Discussion

To our knowledge, this is at present one of the largest retrospective case series investigating the sensitivity, specificity, PPV and NPV, accuracy, and DSC of preoperative CT, cine-MRI, and T1TSE sequence for mediastinal masses. Our series showed excellent NPV and specificity for all investigated structures other than the pericardium. In unclear CT cases, cine-MRI is of greater additional benefit for predicting tumor infiltration into the central thoracic vessels and atria compared with T1TSE sequence. The added value of cine-MRI for aortic infiltration outweighs that of T1TSE sequence significantly.

Preoperative cine-MRI helps to evaluate tumor extent and possible infiltration of adjacent structures. Especially in advanced thymoma and thymic carcinoma (stage III: infiltrating neighboring structures; stage IVA: pleural involvement), infiltration of great intrathoracic vessels or the right atrium is not a contraindication for tumor resection in a multimodality treatment plan [27].

Our MRI protocol is in line with that of other published series and was not changed during the recruitment period. Ong et al. published data for a series of nine patients with preoperative cine-MRI from 2008 to 2011. They used a 1.5 Tesla MRI-scanner (Magnetom Symphony; Siemens Healthcare, Erlangen, Germany) to obtain cardiac-gated T1TSE sequences at a repetition time/echo time (TR/TE) of 558–1194 ms/5.3–7.4 ms with a slice thickness of 6–10 mm and a matrix ranging from 180 × 256 to 216 × 256 [6]. Panda et al. reported a retrospective series of 37 patients from 2010 to 2018 who received cine-MRI on a 1.5 Tesla scanner (Siemens Avanto). They obtained cine-MRI images at repetition time/echo time (TR/TE) of 29–41 ms/1.1–1.3 ms with a slice thickness of 6–8mm and matrix of 192–256 × 208–256 [5]. A slice thickness of ≤5 mm is recommended by the Korean Society of Thoracic Radiology [22].

Ried et al. [28] conducted a prospective study in 2017 investigating tumor infiltration using CT and cine-MRI in the pericardium, myocardium, SVC, and aorta in twelve patients undergoing surgical resection of advanced thymoma or thymic carcinoma. Their sensitivity for pericardial infiltration (CT: 66.7%; cine-MRI: 83.3%) is comparable to our results; however, we found higher rates of sensitivity with CT (CT: 89%; cine-MRI: 69%; T1TSE: 85%). The diagnostic value of CT and cine-MRI (poor sensitivity, specificity, etc.) to predict pericardial infiltration may not be crucial for the decision on operability as pericardial infiltration is usually not a factor technically limiting surgical resection. The pericardium can be surgically resected and replaced with a bovine pericardial patch. Resection of the phrenic nerve may cause severe dyspnea and may require diaphragm plication to restore lung function.

Ong et al. [6] retrospectively analyzed preoperative CT and cine-MRI images of nine patients who underwent surgery. They found a loss of sliding motion in 15 mediastinal structures (e.g., brachiocephalic vein, SVC, ascending aorta, pericardium, and atrium), 14 of which had tumor involvement (adherence or infiltration) at surgery. In six cases (brachiocephalic vein, SVC, aortic arch, descending aorta), tumor involvement was detected intraoperatively but showed normal sliding motion on cine-MRI. They calculated the accuracy, sensitivity, and specificity of mediastinal tumor infiltration for loss of sliding motion on cine-MRI to be 88.5%, 70%, and 97.6%, respectively. They also evaluated tumor infiltration for tumor apposition greater than 90 degrees with loss of intervening fat planes with poorer results. Accuracy, sensitivity, and specificity were 63.9%, 90%, and 51.2%, respectively. Ong et al. recommend cine-MRI especially when ventricular involvement is suspected on CT, as the sliding motion in the ventricles is pronounced [5] [26]. A pericardial effusion impairs the sliding motion as it restricts the mobility of the pericardium [5].

In contrast to Ong et al. [6], Panda et al. [5] examined CSI and cine-MRI retrospectively in 44 patients, 37 of whom underwent surgery and were included in their analysis. They examined 137 cardiovascular structures (e.g., pericardium, aorta, brachiocephalic vein (BCV), pulmonary arteries, etc.) with equivocal CT findings. Accuracy, sensitivity, and specificity were 90.5%, 75.6%, and 96% for cine-MRI and 91.2%, 75.7%, and 97%, respectively.

In their study, the quality of cine-MRI assessment of mediastinal vessels decreased the further away from the heart tumor involvement was suspected. In 75% of cases (9 out of 12 patients) in which the cine-MRI results did not match the intraoperative findings, mediastinal vessels were involved. Large (tumor) masses can reduce sliding motion between two anatomical/tumor structures due to mass effect, leading to false negative cases [5,30].

Between 1997 and 2006, Kajiwara et al. [30] examined 100 lung cancer patients with suspected chest wall infiltration with cine-MRI who underwent surgery for tumor treatment and thus pathological confirmation. In their study, accuracy, sensitivity, and specificity were 47.1%, 60%, and 43.9% for CT and 77%, 100%, and 68.5% for cine-MRI, respectively. Kajiwara et al. found that cine-MRI evaluation is impaired in larger tumors because the weight of the tumor itself limits its mobility.

The most interesting finding of our study is the additional benefit of cine-MRI in unclear CT cases for assessing the infiltration of large thoracic vessels and atria. We were able to correctly predict—in these unclear CT cases—a potential infiltration of the aorta, pulmonary arteries, and atria in 86%, 85%, and 80% of the patients, respectively, using cine-MRI. In contrast, the T1TSE sequence correctly diagnosed aortic infiltration in 29% of patients, pulmonary artery infiltration in 54% of patients, and atrial infiltration in 20% of patients. The additional value of a T1TSE sequence in cases of unclear CT and incorrect cine-MRI patients is very limited—not better than “heads or tails”. Correct T1TSE sequence diagnosis was only achieved in a maximum of 50% of these cases.

Aortic infiltration was significantly more often correctly diagnosed in cine-MRI than T1TSE sequence (*p* = 0.0004). Thus, we highly recommend performing additional cine-MRI in patients where potential infiltration of this structure is suspected. The preoperative surgical treatment strategy may be planned more precisely, thereby improving the patients’ safety. Close cooperation between the thoracic surgeon and the radiologist is of great importance. Performing an additional T1TSE sequence might not provide any extra valuable information, but further stresses the limited resources of the German health care system.

There are some limitations to this study. Due to its retrospective nature, selection bias cannot be eliminated. Another confounding factor is the number of excluded cases due to incomplete data. This was mainly due to missing cine-MRI, T1TSE sequence, and/or CT series. It took a learning curve for adequate imaging acquisition, especially in the early years after cine-MRI implementation. Furthermore, out-of-house CTs sometimes lacked comparable protocols and were not available for re-evaluation in all cases. During the study period (02/2013–01/2022), the head of the thoracic surgery department changed (in 2017); thus, patient selection and indication for surgery as well as a change in guidelines might have also biased our results. Cine-MRI and T1TSE sequence are highly specific imaging modalities requiring excellent radiological expertise and might not be available area-wide. Additional cine-MRI and T1TSE sequence usually do not reflect everyday clinical practice.

## 5. Conclusions

Despite some technical and evaluation limitations, cine-MRI is of major benefit in unclear CT cases and is a useful additional radiologic tool for preoperatively evaluating potential tumor infiltration into large central thoracic vessels and atria. Additionally, T1TSE sequence does not seem to be necessary if aortic infiltration is suspected in cine-MRI. Considering the limited resources in healthcare systems across the world, T1TSE sequence is not necessary and patients may be spared the sometimes-stressful additional imaging modality.

## Figures and Tables

**Figure 1 cancers-16-03162-f001:**
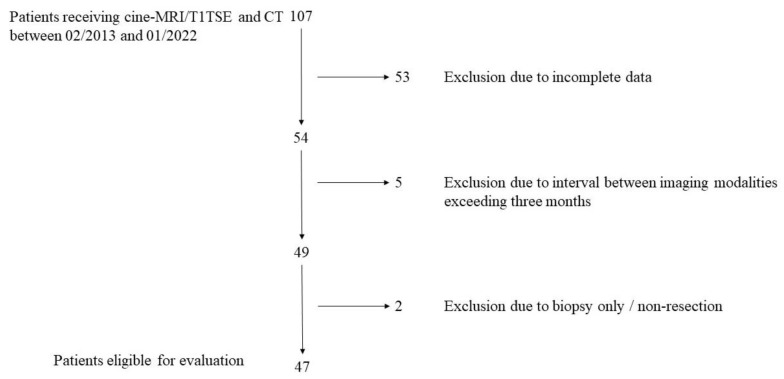
Patient flowchart: flowchart showing patient selection. cine-MRI = cine magnetic resonance imaging, T1TSE = magnetic resonance imaging (MRI)/T1-weighted spin echo sequences, CT = computed tomography.

**Table 1 cancers-16-03162-t001:** Patient characteristics: patient characteristics, surgical approach, and type of resection of the included patients. aTHCT = anterolateral thoracotomy, dTHCT = double thoracotomy, hcTHCT = hemiclamshell thoracotomy, mST = median sternotomy, VATS = video-assisted thoracic surgery, RATS = robotic-assisted thoracic surgery.

Number of Patients		47
Sex		
	Male	18
	Female	29
Age at time of surgery [years]		
	Median	58
	Minimum	22
	Maximum	80
Tumor diameter [cm]		
	Median	6.8
	Minimum	2.2
	Maximum	20.7
Surgical approach		
	aTHCT	12
	dTHCT	5
	hcTHCT	2
	mST	5
	VATS	1
	RATS	22
Type of resection		
	Primary resection	33
	Primary resection after neoadjuvant chemotherapy	11
	Recurrence resection	3

**Table 2 cancers-16-03162-t002:** Final histology reports: final histology reports of included patients. ACC = adenoid cystic carcinoma, SFT = solitary fibrous tumor.

Final Histology	47
Bronchogenic cyst	1
Atypical carcinoid	1
Ectopic goiter	1
Mediastinal germ cell tumor	1
Breast cancer metastasis	1
Osteosarcoma metastasis	1
Neurofibroma	1
Pericardial cyst	1
ACC metastasis	1
Rhabdomyosarcoma metastasis	1
SFT	1
Venous malformation	1
Thymus hyperplasia	8
Thymic cyst	1
Thymic carcinoma	7
Thymoma A	2
Thymoma A/B	5
Thymoma B1	5
Thymoma B2	4
Thymoma B2/B3	1
Thymoma B3	2

**Table 3 cancers-16-03162-t003:** Primary endpoints: synopsis of calculated sensitivity, specificity, positive and negative predictive values, accuracy, and DSC (primary endpoints) for each anatomical structure. If the denominator result is Zero, the value “not possible” is shown. Positive predictive value = PPV, Negative predictive value = NPV, DSC = Dice similarity coefficient, CT = computed tomography, cine-MRI = cine magnetic resonance imaging, T1TSE = magnetic resonance imaging (MRI)/T1-weighted spin echo sequences, SVC = superior vena cava, np = not possible.

	Sensitivity	Specificity	PPV	NPV	Accuracy	DSC
Pericardium						
CT	0.89	0.5	0.4	0.92	0.61	0.55
cine-MRI	0.69	0.64	0.55	0.76	0.66	0.61
T1TSE	0.85	0.83	0.73	0.90	0.83	0.79
Myocardium						
CT	np	0.98	0	1	0.98	0
cine-MRI	np	0.98	0	1	0.98	0
T1TSE	np	0.98	0	1	0.98	0
SVC						
CT	np	0.96	0	1	0.96	0
cine-MRI	0	0.95	0	0.98	0.93	0
T1TSE	np	0.95	0	1	0.95	0
Aorta						
CT	0.5	0.92	0.33	0.96	0.88	0.40
cine-MRI	0.5	0.93	0.25	0.98	0.91	0.33
T1TSE	1	0.86	0.2	1	0.87	0.33
Pulmonary artery						
CT	np	0.94	0	1	0.94	0
cine-MRI	np	0.95	0	1	0.95	0
T1TSE	np	0.95	0	1	0.95	0
Atria						
CT	np	0.86	0	1	0.86	0
cine-MRI	np	0.96	0	1	0.96	0
T1TSE	np	0.93	0	1	0.93	0

**Table 4 cancers-16-03162-t004:** Unclear CT cases further investigated with cine-MRI and T1TSE sequence. Aortic infiltration was significantly more often correctly diagnosed with cine-MRI than with T1TSE sequence (*p* = 0.0004). CT = computed tomography, Cine-MRI = cine magnetic resonance imaging, T1TSE = magnetic resonance imaging (MRI)/T1-weighted spin echo sequences, SVC = superior vena cava.

	CT	Cine-MRI				T1TSE				Fisher’s Exact Test *p*-Value
	Unclear	Correct (True Positive, True Negative)		Incorrect (False Positive, False Negative, Unclear)		Correct (True Positive, True Negative)		Incorrect (False Positive, False Negative, Unclear)		
Pericardium	14	6	43%	8	57%	6	43%	8	57%	1
Myocardium	2	1	50%	1	50%	1	50%	1	50%	1
SVC	2	0	0%	2	100%	0	0%	2	100%	1
Aorta	21	18	86%	3	14%	6	29%	15	71%	0.0004
Pulmonary arteries	13	11	85%	2	15%	7	54%	6	46%	0.2016
Atria	5	4	80%	1	20%	1	20%	4	80%	0.2063

## Data Availability

The (primary) data that support the findings of this study are not publicly available due to their containing information that could comprise the privacy of research participants but are available from the authors Eichinger Monika and Winter Hauke.

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
