# Peer review of "Cine-MRI and T1TSE Sequence for Mediastinal Mass"

_cancers, 2024, doi:10.3390/cancers16183162_

Round 1

Reviewer 1 Report

Comments and Suggestions for Authors

1.       The purpose paragraph of the study contains necessary details regarding the method. A few more concise and understandable sentences expressing the main purposes should be given.

2.       The advantages and disadvantages of mediastinal MRI static sections compared to cine MRI should be given in the introduction. Additionally, it should be detailed in which other mediastinal pathologies cine MRI is used.

3.       The material method section is written very irregularly and complexly. Inclusion and exclusion criteria should be clarified. MRI protocols and CT protocols should be presented more clearly in separate paragraphs. There is no need to explain PPV and NPV; instead, this detail should be given with details of the statistical method used under the heading of statistical analysis.

4.       Were conventional MRI sequences evaluated within the scope of the study? Were you only looking at cine images? This situation should be clarified. The contribution of conventional images to cine should be emphasized. A comparison of these two methods should be made in the discussion section.

5.       The discussion part is too superficial. It should be detailed by adding more up-to-date sources.

6.       Common words in the title and keywords should be replaced with different words.

7.       The results section is designed too ambitiously compared to the study data. It should be revised to include softer sentences.

8.       Red font should not be used in image naming, it contrasts with the image and creates a bad image.

Author Response

Dear Reviewer 1, 

thank you very much for your helpful comments. 

We would like to reply in a Point-by-Point Manner. 

Please see the attached Word file. 

Kind regards

Reviewer 2 Report

Comments and Suggestions for Authors

General comment:

This work deals with cine-MRI for mediastinal tumors infiltrated in aorta and other vessels, comparing this technique to CT and T1TSE methodologies. A cohort of 47 patients was considered. 

Specific comments throughout the paper: 

1. 

The introduction is not focused, not exhaustive and clear enough to provide immediately the knowledge gaps and the novelty of this work. The analysis of the literature is limited and does not provide enough insights. The introduction section must be empowered.

2. 

Line 82: why three months? Tumor progression may vary and this period of time may be enough. 

Lines 83-85: this attempt of non introducing a bias can actually lead to same bias. The authors must support, with quantitative reasoning or references. 

Lines 102-108: please revise the math notation 

About the cine-MRI information, the B0 field strength and resonance freq. are missing, the set of coils used for the subjects (and any eventual differences) are not reported. These lacks are relevant. 

Eq. For PPV, NPV, etc. are not numbered.

3.

There is a mixture of results and methods in this section. The paper must be re-organized. 

Tables are badly formatted. Please revise.

The results presentation is messy and not discussed in details. It must be enhanced in terms of clarity and quality. 

4.

The discussion section is appreciated, even though there is no use of refs. From the literature, thus lowering the quality of the discussion itself. 

5. 

Conclusion section is very short and future perspectives are missing. 

Sect. 6 should be removed. 

Minor issues:

The referencing style in the text is no punctuated, please fix as per the journal’ guidelines. 

Line 102: please thoroughly proofread the manuscript and remove the typos. 

Comments on the Quality of English Language

The english language is relatively fair 

Author Response

Dear Reviewer 2, 

thank you very much for your helpful comments. 

We would like to reply in a Point-by-Point Manner. 

Please see the attached Word file. 

Kind regards

Round 2

Reviewer 1 Report

Comments and Suggestions for Authors

the final version has significantly improved.

Reviewer 2 Report

Comments and Suggestions for Authors

The introduction section has been revised and improved, while new references have been added, thus stressing the need for this article. 

The authors have clarified several methodological details, while revising the organization of the related sections, helping the reader in navigating and reproducing the methods. 

Results presentation is now readable and understandable. 

The discussion section has been improved.

I do not have any further comment.  

Comments on the Quality of English Language

Minor english errors to be checked during proofread.